# How gradient estimator variance and bias could impact learning in neural circuits

**Arna Ghosh**
McGill University &
Mila-Quebec AI Institute
Montréal, QC, Canada
arna.ghosh@mail.mcgill.ca

**Yuhan Helena Liu**
University of Washington
Seattle, WA, USA
Mila-Quebec AI Institute
Montréal, QC, Canada

**Guillaume Lajoie**
Unversite de Montréal &
Mila-Quebec AI Institute
Montréal, QC, Canada

**Konrad Körding**
University of Pennsylvania
Philadelphia, PA, USA
CIFAR Learning in Machines & Brains
Toronto, ON, Canada

**Blake A. Richards**
McGill University, Mila-Quebec AI Institute &
Montreal Neurological Institute
Montréal, QC, Canada
CIFAR Learning in Machines & Brains
Toronto, ON, Canada
blake.richards@mcgill.ca

## Abstract

There is growing interest in understanding how real brains may approximate gradients and how gradients can be used to train neuromorphic chips. However, neither real brains nor neuromorphic chips can perfectly follow the loss gradient, so parameter updates would necessarily use gradient estimators that have some variance and/or bias. Therefore, there is a need to understand better how variance and bias in gradient estimators impact learning dependent on network and task properties. Here, we show that variance and bias can impair learning on the training data, but some degree of variance and bias in a gradient estimator can be beneficial for generalization. We find that the ideal amount of variance and bias in a gradient estimator are dependent on several properties of the network and task: the size and activity sparsity of the network, the norm of the gradient, and the curvature of the loss landscape. As such, whether considering biologically-plausible learning algorithms or algorithms for training neuromorphic chips, researchers can analyze these properties to determine whether their approximation to gradient descent will be effective for learning given their network and task properties.

## 1 Introduction

Artificial neural networks (ANNs) typically use gradient descent and its variants to update their parameters in order to optimize a loss function (LeCun et al., 2015; Rumelhart et al., 1986). Importantly, gradient descent works well, in part, because when making small updates to the parameters, the loss function's gradient is along the direction of greatest reduction.[1] Motivated by these facts, a longstanding question in computational neuroscience is, does the brain approximate gradient descent (Lillicrap et al., 2020; Whittington & Bogacz, 2019)? Over the last few years, many papers show that, in principle, the brain could approximate gradients of some loss function (Murray, 2019; Liu et al., 2021; Payeur et al., 2021; Lillicrap et al., 2016; Scellier & Bengio, 2017). Also inspired by the brain, neuromorphic computing has engineered unique materials and circuits that emulate biological networks in order to improve efficiency of computation (Roy et al., 2019; Li et al., 2018b). But, unlike ANNs, both real neural circuits and neuromorphic chips must rely on approximations to the true gradient. This is due to noise in biological synapses and memristors, non-differentiable operations such as spiking, and the requirement for weight updates that do not use non-local information (which can lead to bias) (Cramer Benjamin et al., 2022; M. Payvand et al., 2020b; Laborieux et al., 2021; Shimizu et al., 2021; Neftci et al., 2017; N. R. Shanbhag et al., 2019). Thus, both areas of research

---

[1]If we assume a Euclidean metric in weight space.

could benefit from a principled analysis of how learning is impacted by loss gradient variance and bias.

In this work, we ask how different amounts of noise and/or bias in estimates of the loss gradient affect learning performance. As shown in a simple example in Fig. 9, learning performance can be insensitive to some degree of variance and bias in the gradient estimate, and even benefit from it, but excessive amounts of variance and/or bias clearly hinder learning performance. Results from optimization theory shed light on why imperfectly following the gradient—e.g. via stochastic gradient descent (SGD) or other noisy GD settings—can improve generalization in ANNs (Foret et al., 2020; Chaudhari et al., 2019; Yao et al., 2018; Ghorbani et al., 2019). However, most of these results treat unbiased gradient estimators. In contrast, in this work, we are concerned with the specific case of weight updates with intrinsic but known variance and bias, as is often the case in computational neuroscience and neuromorphic engineering. Moreover, we also examine how variance and bias can *hinder* training, because the amount of variance and bias in biologically-plausible and neuromorphic learning algorithms is often at levels that impair, rather than improve, learning (Laborieux et al., 2021), and sits in a different regime than that typically considered in optimization theory.

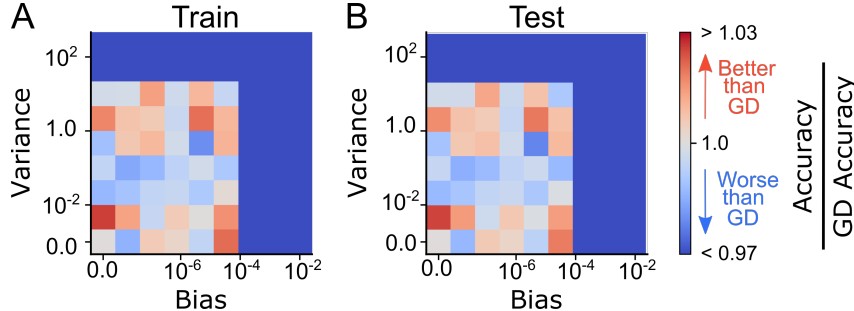

Figure 1: Train and test accuracy of a VGG-16 network trained for 50 epochs (to convergence) on CIFAR-10 using full-batch gradient descent (with no learning rate schedule) with varying amount of variance and bias (as a fraction of the gradient norm) added to the gradient estimates. These results (avg of 20 seeds) indicate that excessive noise and bias harms learning, but a small amount can aid it.

The observations in Fig. 9 give rise to an important question for computational neuroscientists and neuromorphic chip designers alike: what amount of variance and bias in a loss gradient estimate is tolerable, or even desirable? To answer this question, we first observe how variance and bias in gradient approximations impact the loss function in a single parameter update step on the training data. (We also extend this to multiple updates in Appendix A.) We utilize an analytical and empirical framework that is agnostic to the actual learning rule, and derive the factors that affect performance in an imperfect gradient setting. Specifically, we assume that each update is comprised of the contribution of the true gradient of the loss function with respect to the parameter, a fixed amount of bias, and some noise. Similar to Raman et al. (2019), we derive an expression for the change in the loss function after a discrete update step in parameter space: $w(t + \Delta t) = w(t) + \Delta w(t)\Delta t$, where $\Delta t$ is akin to learning rate in standard gradient descent algorithms. We then characterize the impact on learning using the decrease in loss function under an approximate gradient setting as compared to the decrease in loss function when following the true gradient.

Our analysis demonstrates that the impact of variance and bias are independent of each other. Furthermore, we empirically validate our inferences in ANNs, both toy networks, and various VGG configurations (Vedaldi & Zisserman, 2016) trained on CIFAR-10 (Krizhevsky & Hinton, 2009). Our findings can be summarized as follows:

1. The impact of variance is second order in nature. It is lower for networks with a threshold non-linearity, as compared to parameter matched linear networks. Under specific conditions, variance matters lesser for wider and deeper networks.

2. The impact of bias increases linearly with the norm of the gradient of the loss function, and depends on the direction of the gradient as well as the eigenvectors of the loss Hessian.

3. Both variance and bias can help to prevent the system from converging to sharp minima, which may improve generalization performance.

Altogether, our results provide guidelines for what network and task properties computational neuroscientists and neuromorphic engineers need to consider when designing and using noisy and/or biased gradient estimators for learning.

## 2 ESTIMATING THE IMPACT OF VARIANCE AND BIAS ON TRAINING

Our analysis focuses on situations where the synaptic weights of a network, $w$, are trained to optimize a loss, $\mathcal{L}[\boldsymbol{w}]$. We use an auxiliary variable, $t$, to denote different points in this optimization trajectory and $\Delta t$ to denote a single step in this trajectory. (We extend to the multi-step case in Corollary A.2 of the Appendix.) Compared to standard optimization protocols in the ANN literature, $\Delta t$ is akin to the learning rate. Therefore, the loss as measured at one particular point in the trajectory is denoted as $\mathcal{L}[\boldsymbol{w}(t)]$, and the loss at the next point in the trajectory, i.e., after a weight update step, is denoted by $\mathcal{L}[\boldsymbol{w}(t + \Delta t)]$. In the rest of this work, we characterize $\mathcal{L}[\boldsymbol{w}(t + \Delta t)] - \mathcal{L}[\boldsymbol{w}(t)]$ as a function of the variance and bias in the weight update.

We assume that $\mathcal{L}$ is twice differentiable and $\Delta t$ is small enough such that $\mathcal{L}[\boldsymbol{w}(t + \Delta t)]$ can be effectively approximated around $\mathcal{L}[\boldsymbol{w}(t)]$ using a second order Taylor series, i.e., higher order terms beyond the second order can be ignored. Furthermore, we define a weight update equation that consists of the gradient, a fixed bias, and some noise. We denote the bias vector as $b\overrightarrow{\boldsymbol{\beta}}$, where $\overrightarrow{\boldsymbol{\beta}}$ is a norm $\sqrt{N}$ vector that indicates the direction of bias in the $N$-dimensional parameter space. A special case of $\overrightarrow{\boldsymbol{\beta}}$ would be $\overrightarrow{\boldsymbol{\beta}} = \overrightarrow{\boldsymbol{1}}$ when all parameters have the same bias. Similarly, we assume white noise in the gradient estimates. With these assumptions, we define the weight update equation to be:

$$\Delta w(t) := -\nabla_{\boldsymbol{w}}\mathcal{L}[\boldsymbol{w}(t)] + b\overrightarrow{\boldsymbol{\beta}} + \sigma\sqrt{f(N)}\hat{n} \tag{1}$$

where $\nabla_{\boldsymbol{w}}\mathcal{L}[\boldsymbol{w}(t)]$ denotes the first derivative of the gradient of $\mathcal{L}$ evaluated at $w(t)$ and $\hat{n}$ is a unit vector in the $N$-dimensional parameter space whose elements are zero-mean i.i.d. (see Appendix A for generality of this assumption), such that the total variance of $\Delta w(t)$ is $\sigma^2 f(N)$, where $f(N)$ denotes the dependence of variance on the total number of network parameters, $N$. Finally, we define $\Delta\mathcal{L}_t(b, \sigma^2)$ as the change in $\mathcal{L}[\boldsymbol{w}(t)]$ when performing the aforementioned weight update as compared to the change in $\mathcal{L}[\boldsymbol{w}(t)]$ when the weight update step follows the true gradient, i.e.,

$$\Delta\mathcal{L}_t(b, \sigma^2) = [\mathcal{L}[\boldsymbol{w}(t + \Delta t)] - \mathcal{L}[\boldsymbol{w}(t)]]_{(b,\sigma^2)} - [\mathcal{L}[\boldsymbol{w}(t + \Delta t)] - \mathcal{L}[\boldsymbol{w}(t)]]_{(0,0)} \tag{2}$$

where $[\mathcal{L}[\boldsymbol{w}(t + \Delta t)] - \mathcal{L}[\boldsymbol{w}(t)]]_{(b,\sigma^2)}$ is the change in loss with bias $b$ and variance $\sigma^2$. For brevity, we drop the $t$ is the subscript in the rest of the paper. Formally, we present the following Lemma:

**Lemma 2.1.** *Second order Taylor series expansion. Assuming a small learning rate, the bias, $b$, and variance, $\sigma^2$, in gradient estimates can be linked to changes in $\mathcal{L}$ upon one weight update step as compared to the change in $\mathcal{L}$ under a true gradient descent weight update step:*

$$\mathbb{E}_{\hat{n}}\left[\Delta\mathcal{L}(b, \sigma^2)\right] = \mathbb{E}_{\hat{n}}\left[[\mathcal{L}[\boldsymbol{w}(t + \Delta t)] - \mathcal{L}[\boldsymbol{w}(t)]]_{(b,\sigma^2)} - [\mathcal{L}[\boldsymbol{w}(t + \Delta t)] - \mathcal{L}[\boldsymbol{w}(t)]]_{(0,0)}\right]$$

$$= b\left\langle\nabla_{\boldsymbol{w}}\mathcal{L}[\boldsymbol{w}(t)], \overrightarrow{\boldsymbol{\beta}}\right\rangle\Delta t + \frac{1}{2}b^2\left\langle\overrightarrow{\boldsymbol{\beta}}, \nabla_{\boldsymbol{w}}^2\mathcal{L}[\boldsymbol{w}(t)]\overrightarrow{\boldsymbol{\beta}}\right\rangle\Delta t^2$$

$$-\frac{1}{2}b\left\langle\nabla_{\boldsymbol{w}}\mathcal{L}[\boldsymbol{w}(t)], (\nabla_{\boldsymbol{w}}^2\mathcal{L}[\boldsymbol{w}(t)] + \nabla_{\boldsymbol{w}}^2\mathcal{L}[\boldsymbol{w}(t)]^T)\overrightarrow{\boldsymbol{\beta}}\right\rangle\Delta t^2 + \frac{1}{2}\frac{\sigma^2 f(N)}{N}\text{Tr}[\nabla_{\boldsymbol{w}}^2\mathcal{L}[\boldsymbol{w}(t)]]\Delta t^2 \tag{3}$$

*where $\langle\cdot,\cdot\rangle$ denotes the dot product, $\text{Tr}$ denotes the Trace operator of a square matrix, orange terms indicate the impact of bias, and the green term indicates the impact of variance.*

In the next section, we present an in-depth analysis of the impact of variance when $f(N)$ is sublinear, as is the case for some learning algorithms, e.g. Equilibrium Propagation (Laborieux et al., 2021). We present a more general analysis along with the proofs of all lemmas and theorems in Appendix A.

## 2.1 ANALYSIS OF THE IMPACT OF VARIANCE AND BIAS ON TRAINING LOSS

One of the corollaries that follow from Lemma 2.1 is that the impact of variance and bias on the expected $\Delta\mathcal{L}_t(b, \sigma^2)$ are independent of each other. Moreover, it is clear that the impact of variance is directly proportional to the trace of $\nabla_{\boldsymbol{w}}^2 \mathcal{L}[\boldsymbol{w}(t)]$, i.e., the loss Hessian, and inversely proportional to $N$, i.e., the number of trainable parameters in the network.

Previous work investigating the object categorization loss landscape for feedforward ANNs demonstrated a pronounced effect of width, i.e., increasing width leads to smoother loss landscapes (Li et al., 2018a), thereby leading to lower trace of Hessian (Hochreiter & Schmidhuber, 1994). This empirical observation in deep ANNs implies that increasing the width both lowers the trace of the Hessian in addition to its obviously increasing the number of trainable network parameters. Therefore, we can say directly from Lemma 2.1 that the impact of variance is lower for wider networks. However, the role of depth is more complicated. Notably, increasing depth (with no skip connections) could theoretically make the loss landscape less smooth (Li et al., 2018a). Thus, we begin by analyzing the case of increasing depth in linear feedforward networks. We prove that, in fact, increasing depth leads to lower values for $\frac{\text{Tr}[\nabla_{\boldsymbol{w}}^2 \mathcal{L}]}{N}$, thereby implying a lower impact of variance.

**Theorem 2.2.** ***Increasing depth lowers impact of variance*** *For linear feedforward ANNs, the impact of variance on $\Delta\mathcal{L}_t(0, \sigma^2)$ for a $(L+1)$ layer network is less than that of a $L$ layer network, i.e.,*

$$\Delta\mathcal{L}_t(0, \sigma^2)_{(L+1)-layer} \leq \Delta\mathcal{L}(0, \sigma^2)_{L-layer} \tag{4}$$

*where each layer has $d$ units and weights initialized by the Xavier initialization strategy and the total variance in gradient estimates does not grow with the size of the network, i.e., $f(N)$ is some constant[2]*

Although these assumptions on layer weights may not strictly hold over the course of training, empirical experience suggests that there exists a loose correspondence among these quantities if the conditions hold true during initialization. Taken together, overparameterization in either width or depth leads to a lower impact of variance in gradient estimates on the network's training.

Besides the network architecture, the non-linearity used in the network also impacts the function implemented by the network, and therefore affects the trace of the loss Hessian. Specifically, we demonstrate that the trace of the loss Hessian is lower for networks with a threshold non-linearity with gain less than or equal to 1, such as a ReLU operation, as compared to a linear network. Formally,

**Theorem 2.3.** ***ReLU lowers impact of variance*** *Let $\phi(.)$ denote a threshold non-linearity function. The impact of variance on $\Delta\mathcal{L}_t(0, \sigma^2)$ for a network with such non-linearities is less than that of an equivalent linear network, i.e.,*

$$\Delta\mathcal{L}_t(0, \sigma^2)_\phi \leq \Delta\mathcal{L}_t(0, \sigma^2)_{Linear} \tag{5}$$

*where the gain of $\phi(.)$ is less than or equal to 1 (e.g. ReLU).*

Summarily, the standard practices in deep neural networks, like overparameterized networks or the use of ReLU as a non-linearity, lead to a lower impact of variance on the network's training. Interestingly, brains are also arguably overparameterized and characterized by threshold non-linearities that lead to sparse activations. These properties could offer a potential solution to countering any negative impact of variance in gradient estimates in the brain and allow biological agents to reliably train complex tasks despite not being able to actually conduct true gradient descent (Lillicrap et al., 2020). It also suggests neuromorphic chips could tolerate more noise in gradient estimators if they are made larger.

In contrast to the impact of variance, the impact of bias is more nuanced and depends on the norm and direction of gradient, the direction of the bias vector, as well as the eigenvectors of the loss Hessian. Using a few algebraic manipulations, Lemma 2.1 can be extended to show that:

**Theorem 2.4.** *The impact of bias on $\Delta\mathcal{L}_t(b, 0)$ grows linearly with the norm of the gradient.*

$$\Delta\mathcal{L}_t(b, 0) = \frac{1}{2}b^2 Q\Delta t^2 + b\|\nabla_{\boldsymbol{w}}\mathcal{L}[\boldsymbol{w}(t)]\|P\Delta t \tag{6}$$

*where $P, Q$ are terms that are independent of the norm of the gradient, $\|\nabla_{\boldsymbol{w}}\mathcal{L}[\boldsymbol{w}(t)]\|$.*

---

[2]Check Appendix A for an extension to the general case where variance changes with number of parameters.

This result implies that initialization and architectural considerations could play important roles in mitigating the impact of bias on training by limiting the norm of the gradient. In the next section, we will empirically validate these results in linear and shallow networks trained on toy datasets as well as in VGG networks trained on CIFAR-10.

## 2.2 EMPIRICAL VERIFICATION OF THE VARIANCE AND BIAS RESULTS

Our experimental setup (illustrated in Fig. 2) is motivated by Raman et al. (2019), and follows the standard student-teacher framework. A "teacher" ANN is initialized with fixed parameters. The task is for a randomly initialized "student" network to learn the input-output mapping defined by the teacher. We defer the reader to the Appendix B for experimental details.

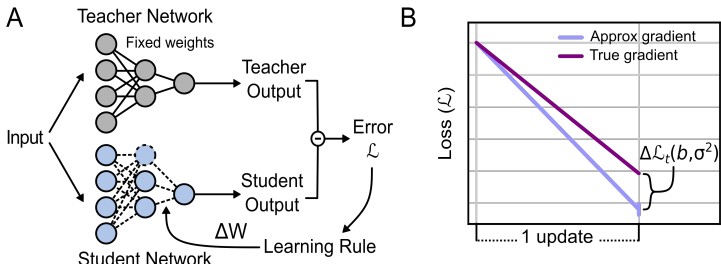

Figure 2: Experimental setup for empirical validation of our results, focusing on $\Delta \mathcal{L}_t(b, \sigma^2)$ to understand how gradient approximations impact learning in one update step. (A) The student-teacher framework motivated by Raman et al. (2019). (B) Illustration of $\Delta \mathcal{L}_t(b, \sigma^2)$ (defined in Eq. (2))

First, we validate our analytical results pertaining to the impact of variance, i.e., Theorems 2.2 and 2.3 in relatively shallow (1-8 hidden layers) fully connected ANNs. We fix the teacher network architecture and use a ReLU non-linearity to threshold the intermediate layer activations. Subsequently, we vary the depth and width of the student network with no non-linearity and observe $\Delta \mathcal{L}_t(0, \sigma^2)$ at different points in the loss landscape. For each such setting, we add a ReLU non-linearity to the student network to validate the effect of a threshold non-linearity on $\Delta \mathcal{L}_t(0, \sigma^2)$. We plot the mean $\Delta \mathcal{L}_t(0, \sigma^2)$ across different points in the loss landscape in Fig. 3. Note that in order to use a sample accurately reflecting the loss landscape that would be encountered during a learning trajectory, we train a control network with the same architecture and non-linearity as the student network using the true gradient and evaluate $\Delta \mathcal{L}_t(0, \sigma^2)$ for each update step along the learning trajectory. In doing so, we are able to observe the desired quantities across a wide range of gradient norm values.

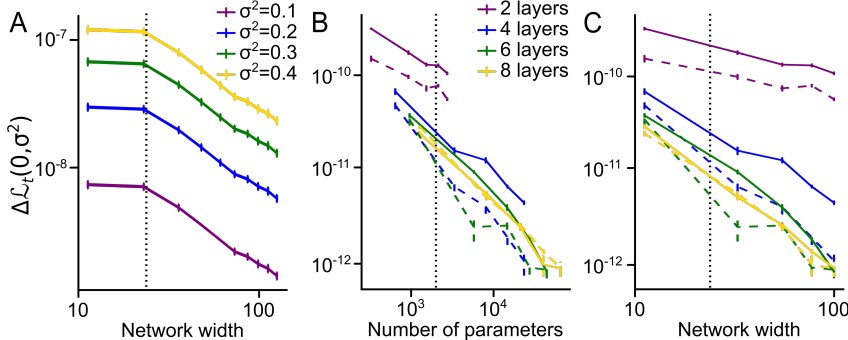

Figure 3: Impact of variance is lower for wider, deeper, and ReLU networks. Solid lines indicate linear networks, dashed lines indicate networks with ReLU non-linearity, and dotted line indicate the teacher network architecture. (A) Varying hidden layer width in one hidden layer linear ANNs, keeping input/output dimensions same. (B) Linear and ReLU networks with varying width and depth plotted with number of parameters in x-axis. (C) Same as B but plotted with network width in x-axis.

Furthermore, we validate Theorems 2.2 and 2.3 on VGG networks trained on the CIFAR-10 dataset. Specifically, we maintain the student-teacher framework and fix the teacher network to be a VGG-19 network trained on CIFAR-10 to 92.6% test accuracy, and we use different networks from the VGG family as student networks (with variance set to $\sigma^2 = 20$). For each network, we use both random weight initialization and Imagenet-pretrained weights to determine whether initialization has an effect on the impact of variance. Fig. 5 demonstrates that our results pertaining to depth and width hold for deep convolutional neural networks (CNNs).

Similarly, we validate Theorem 2.4, first in shallow fully connected ANNs and subsequently in deep CNNs trained on CIFAR-10. Specifically, we demonstrate in Fig. 4 that the impact of bias (with $b = 0.02$) on performance of a linear shallow network grows with the norm of the gradient. Note that the sign duality of $\Delta \mathcal{L}_t(b, 0)$ in this figure follows from Theorem 2.4 where the quantities $P$ and $Q$ depend on the direction of the bias vector with respect to the direction of gradient and the eigenvectors of the loss Hessian (see proof in Appendix A for more details). Therefore, bias can help or hinder learning depending on its direction. This behaviour is in contrast to variance, which always hinders learning for a convex loss (Lemma 2.1). Nonetheless, it is worth noting that when the gradient norm is small as bias increases it always hinders training (see Fig. 4B).

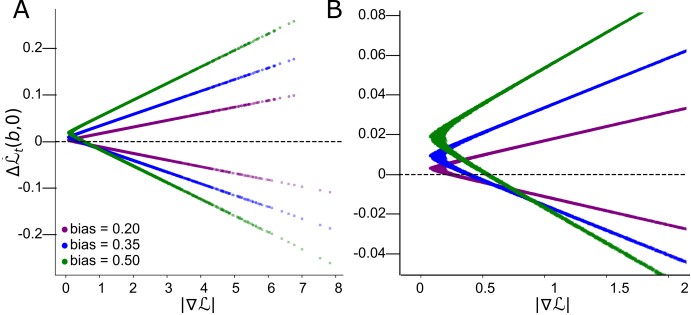

Figure 4: Impact of bias grows linearly with gradient norm and has a quadratic relationship with the amount of bias when the gradient norm tends to 0, i.e., validating expression from Theorem 2.4: $\Delta \mathcal{L}_t(b, 0) = \frac{1}{2} b^2 Q \Delta t^2 + b \|\nabla_{\boldsymbol{w}} \mathcal{L}[\boldsymbol{w}(t)]\| P \Delta t$. (A) Impact of bias grows linearly with gradient norm, with the slope being the amount of bias (see second term in equation). (B) Impact of bias grows quadratically with amount of bias when gradient norm is small (see first term in equation).

Following this, we measure the absolute value of $\Delta \mathcal{L}_t(b, 0)$ and plot its mean across multiple update steps in VGG networks trained on CIFAR-10. In Fig. 6, we show that the relation stated in Theorem 2.4 holds across different VGG architectures, irrespective of the weight initialization. Furthermore, deeper VGG networks have a lower impact of bias owing to a lower gradient norm.

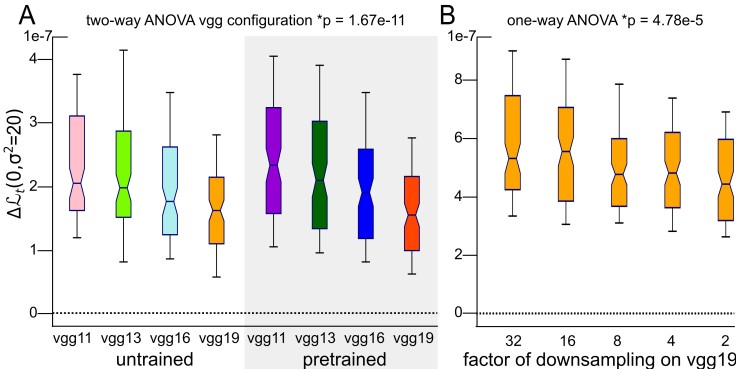

Figure 5: Impact of variance is lower for deeper and wider VGG networks when training on CIFAR-10. (A) Deeper VGG configurations, both Imagenet-pretrained or untrained, exhibit lower impact of variance. (B) Wider VGG-19 configuration networks exhibit lower impact of variance. Factor of downsampling indicates the reduction in the number of filters in convolutional layers and dimensionality of intermediate fully connected layers compared to the original VGG-19 configuration.

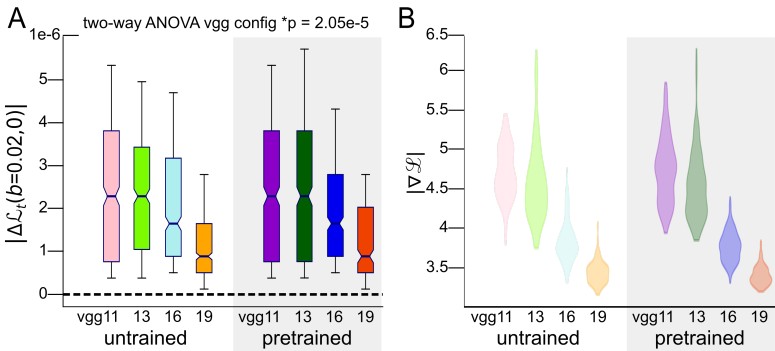

Figure 6: Impact of bias is lower for (A) deeper VGG networks when training on CIFAR-10. (B) Deeper VGG networks exhibit lower gradient norm, thereby mitigating the impact of bias in training.

## 3 ESTIMATING THE IMPACT OF VARIANCE AND BIAS ON GENERALIZATION

In the previous section, we studied how variance and bias in gradient estimates impacts training. However, from a machine learning perspective, an important question is to understand the impact on the system's generalization. Specifically, we ask the question: under what conditions could variance and bias in gradient estimates aid generalization performance? Interestingly, current deep learning practices rely on not following the exact gradient of the loss function to train models, which have been demonstrated to help generalization performance. Some common examples of such practices include stochastic gradient descent (SGD) and dropout[3].

In this section, we use our framework to understand how variance and bias in gradient estimates could alter the learning trajectory, thereby impacting generalization. Specifically, we investigate the conditions under which a parameter update would necessarily lead to descent on the loss landscape. This matters for understanding generalization because the flatness of the loss landscape is a good proxy for generalization (Baldassi et al., 2020; Jiang et al., 2019; Sankar et al., 2021; Tsuzuku et al., 2020; Petzka et al., 2021): flatter minima tend to have better generalization performance than sharper ones. We leverage this viewpoint to understand the impact of variance and bias on generalization. Specifically, we investigate the conditions under which gradient approximations could help to avoid sharp minima, thereby promoting convergence to wide flat minima.

### 3.1 ANALYSIS OF THE IMPACT OF VARIANCE AND BIAS DESCENT OF NARROW MINIMA

To quantify the flatness of loss landscape, we use eigenvalues of the loss Hessian. Without loss of generality, we assume that the loss Hessian is a normal square matrix, i.e., its eigenvectors form an orthogonal basis of the $N$-dimensional parameter space. In order to understand the properties of loss minima that the network could potentially converge to, we derive the sufficiency conditions under which a parameter update ascends (or descends) the loss landscape. This strategy indicates that noise in gradient estimates would lead to the system not descending the loss landscape when the minima is too sharp, which can aid generalization.

**Theorem 3.1.** ***Noise helps avoid sharp minima*** *The sufficient condition under which noise prevents the system from descending the loss landscape, i.e., $\mathcal{L}[\boldsymbol{w}(t + \Delta t)] \geq \mathcal{L}[\boldsymbol{w}(t)]$, is*

$$\lambda_1 \geq \lambda_N \geq \frac{2}{\Delta t} \frac{1}{1 + \left(\frac{\sigma}{\|\nabla_{\boldsymbol{w}}\mathcal{L}\|}\right)^2} \tag{7}$$

*where $\lambda_1 \geq \lambda_2... \geq \lambda_N$ denote the eigenvalues of $\nabla_{\boldsymbol{w}}^2\mathcal{L}$ around the minima.*

Interestingly, the condition presented in Theorem 3.1 presents a lower bound for the smallest eigenvalue, i.e., the eigenvalue associated with the direction of least curvature, and connects it to the variance to gradient norm ratio. This ratio can be thought of as the inverse noise ratio when there is

---

[3]Discussion on the relationship between our analysis and SGD and dropout in the Appendix C.2

no bias in the approximation. Furthermore, we also show in Appendix A that the sufficient condition for descending the loss landscape is similar to the one presented in Theorem 3.1, but provides an upper bound for the largest eigenvalue, i.e., $\lambda_1 \leq \frac{2}{\Delta t} \frac{1}{1+\left(\frac{\sigma}{\|\nabla_{\boldsymbol{w}}\mathcal{L}\|}\right)^2}$. Note that as we get closer to the minimum, the gradient norm would decrease (property of smooth $\mathcal{L}$), and therefore, the inverse noise ratio would increase. Thus, the upper bound on curvature of loss minima that the system can potentially converge to decreases. Taken together, this shows that variance helps the system avoid converging to sharp minima and instead promotes converging to wide flat minima.

Bias, on the other hand, has a more nuanced effect on the learning trajectory. This complexity is unsurprising given the dependence of the impact of bias on direction of the bias vector, the direction of the gradient, and the eigenvectors of the loss Hessian. Once again, we derive the sufficiency condition for when a parameter update step would produce a decrease in the loss function. It turns out this condition depends on both minima flatness and amount of bias relative to the gradient norm.

**Theorem 3.2.** *Bias prevents descent into local minima dependent on Hessian spectrum. Bias in gradient estimates could prevent converging to a minima. Specifically, the sufficient condition for the weight updates to produce a decrease in $\mathcal{L}$ when using a biased estimate of the gradient is:*

$$\frac{b}{\|\nabla_{\boldsymbol{w}}\mathcal{L}\|} \leq \frac{1}{\sqrt{N}} \frac{2}{1+\psi\Delta t + \sqrt{1+4\psi\Delta t + \psi^2\Delta t^2}} \approx \frac{1}{\sqrt{N}} \frac{1}{\left(1+\frac{3}{2}\psi\Delta t\right)} \tag{8}$$

*where $\psi^2 = \sum_i \lambda_i^2$, i.e., the Frobenius norm of the loss Hessian, and $\lambda_1 \geq \lambda_2 ... \geq \lambda_N$ denote the eigenvalues of $\nabla_{\boldsymbol{w}}^2\mathcal{L}$ around the minima and $\nabla_{\boldsymbol{w}}^2\mathcal{L}$ is a normal matrix.*

The condition in Theorem 3.2 indicates that a system that follows a biased gradient approximation will descend the loss landscape until some neighbourhood around a minimum (characterized by the minima flatness and relative bias). Inside this neighbourhood, the system may not descend the loss landscape and therefore not converge to the minimum. The condition depends on minima flatness due to the $\psi$ term (Eq. (8)): sharper (higher curvature) minima should have higher $\psi$, which makes it less likely to satisfy the condition. The condition also depends on the amount of bias relative to gradient norm, i.e.,, higher relative bias makes it less likely to satisfy the condition for descent, which we also support through simulation (see Fig. 7C). Taken together, variance and bias in gradient estimates promote gradient descent dynamics that converge to wide flat minima, which can aid generalization. This effect is empirically demonstrated by our first example of VGG-16 networks trained on CIFAR-10 (Fig. 9).

## 3.2 EMPIRICAL VERIFICATION OF THE IMPACT OF VARIANCE AND BIAS ON GENERALIZATION

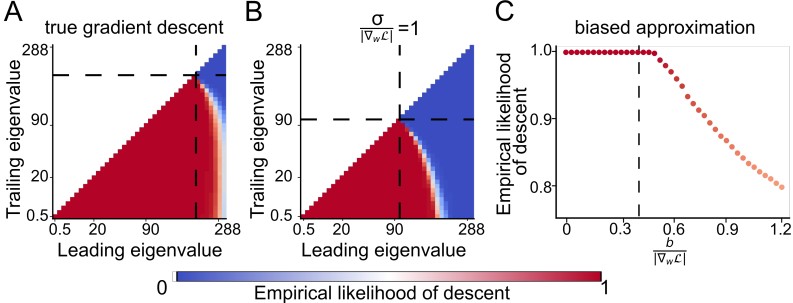

Figure 7: Empirical verification of Theorems 3.1 and 3.2 in linear ANNs. Dashed lines indicate the theoretical limit and colours indicate the empirical probability of descent. (A) True gradient descent. (B) Noise in gradient estimates, $\sigma = \|\nabla_{\boldsymbol{w}}\mathcal{L}\|$. (C) Bias in gradient estimates, ratio $\frac{b}{\|\nabla_{\boldsymbol{w}}\mathcal{L}\|}$ is varied.

We follow the same experimental setting as described in Section 2.2 for verifying Theorems 3.1 and 3.2. Owing to computational bottlenecks in loss Hessian estimation for high-dimensional nonlinear settings, we restrict our empirical validation to training linear networks for optimizing MSE loss. In doing so, we gain full control over the loss Hessian via the input covariance statistics[4]— Fig. 7A & B demonstrate that our inferences from Theorem 3.1 hold empirically. When the leading

---

[4]for MSE loss at minima, the Hessian is equal to the input covariance matrix barring a constant factor

eigenvalue is *less than* the theoretical limit, a weight update step always leads to a decrease in $\mathcal{L}$; when the trailing eigenvalue is *greater* than the theoretical limit, a weight update step always leads to an increase in $\mathcal{L}$. Furthermore, the theoretical limit is higher for the case when there is no variance than with variance. These observations demonstrate that following a noisy version of the gradient helps the network avoid sharp local minima. Similarly, Fig. 7C demonstrates that our inferences from Theorem 3.2 hold empirically. Specifically, when the ratio of bias to gradient norm is below the theoretical threshold, a weight update step always leads to a decrease in $\mathcal{L}$.

## 4    DISCUSSION

In both computational neuroscience and neuromorphic computing, gradient estimators are known to have variance and bias due to both algorithmic and hardware constraints (Laborieux et al., 2021; M. Payvand et al., 2020a; Lillicrap et al., 2020; Pozzi et al., 2018; Sacramento et al., 2018; Rubin et al., 2021; Roelfsema & Holtmaat, 2018; Bellec et al., 2020; Neftci et al., 2019; Huh & Sejnowski, 2018; Zenke & Neftci, 2021). Using mathematical analysis and empirical verification, we found that the impact of variance and bias on learning is determined by the network size, activation function, gradient norm, and loss Hessian, such that the amount of variance and bias that are permissible or even desirable depends on these properties. Though our empirical verification was done using feedforward ANNs, our mathematical analysis was formulated to be as general as possible, and we believe that our assumptions are reasonable for most cases that computational neuroscientists and neuromorphic chip engineers face. Furthermore, we add a corollary to Lemma 2.1 and discuss extensions of our analysis to multi-step optimization settings in Appendix A (see Corollary A.2). Thus, our work can inform research in these areas by providing a guide for how much variance and bias relative to the gradient norm is reasonable for a given network.

### 4.1    RELATED WORK

Our analysis was inspired in part by recent work by Raman et al. (2019) characterizing bounds on learning performance in neural circuits. They demonstrated that in the absence of noise in the gradient estimator, larger networks are always better at training, but with noise added, there is a size limit beyond which training is impaired. Our work was also informed by research into generalization in deep ANNs, which provided the rationale for our analyses examining the potential for a learning algorithm to descend into sharp minima or not (Foret et al., 2020; Chaudhari et al., 2019; Yao et al., 2018; Ghorbani et al., 2019; Smith et al., 2021). As well, our work has some clear relationship to other work examining the variance of gradient estimators (e.g. Werfel et al. (2003)), but here, we are asking a novel question, namely, how does a given amount of variance and bias impact performance for different network parameters?

More broadly, our work relates to the deep learning literature because modern ANNs rarely use the true loss gradient over the entire dataset to make weight updates. Instead, it is common practice to add noise in different forms, e.g. SGD (Bottou, 2012), dropout (Srivastava et al., 2014), dropconnect (Wan et al., 2013) and label noise (Blanc et al., 2020; HaoChen et al., 2021; Damian et al., 2021). But, the noise structure in these scenarios may differ from assumptions in our work. For example, SGD does not exhibit white noise structure (Xie et al., 2021; Zhu et al., 2019). [5] Likewise, dropout can be thought of as adding noise, but for a truly unbiased estimate of the gradient, one would have to consider all possible configurations of dropped neurons, something quite uncommon in practice.

### 4.2    LIMITATIONS AND FUTURE WORK

This work focused on characterizing the impact of variance and bias in gradient estimates, but it lacks any in-depth analysis of specific bio-plausible or neuromorphic learning algorithms. Similarly, our work only characterizes learning performance, but does not provide a concrete proposal to mitigate excessive noise or bias in gradient approximations. As such, our analyses can be used by other researchers to assess their learning algorithms but they do not directly speak to any specific learning algorithm nor provide mechanisms for improving existing algorithms. Nonetheless, we believe that our work provides a framework to help develop such strategies, and we leave it to future work.

---

[5] Though several analytical studies have used white noise to model SGD dynamics, and concluded that the SGD noise acts as a regularizer against converging to sharp minima (Li et al., 2021), similar to our analyses.

## REPRODUCIBILITY STATEMENT

We strongly believe that reproducibility is critical for research progress in both machine learning and computational neuroscience. To this end, we have provided thorough experimental details in the appendix, and in the supplementary materials, we have included the code to run all of the experiments and generate the figures. Our code can also be accessed from the project's github repo. We believe that this information will allow the community to validate/replicate our results and further build on our work.

## ACKNOWLEDGEMENTS

The authors would like to thank Abhishek Banerjee, Jonathan Cornford, and Shahab Bakhtiari for insightful discussions and helpful feedback over the course of the project, as well as Zahraa Chorghay and Colleen Gillon for their aesthetic contribution to the manuscript and figures. This research was generously supported by Vanier Canada Graduate scholarship and Healthy Brains for Healthy Lives fellowship (A.G.); NSERC PGS-D and NSF AccelNet IN-BIC program, Grant No. OISE-2019976 AM02 (Y.H.L.); NSERC Discovery Grant RGPIN-2018-04821, FRQS Research Scholar Award, Junior 1 LAJGU0401-253188, Canada Research Chair in Neural Computations and Interfacing (G.L.); NSERC (Discovery Grant: RGPIN-2020-05105; Discovery Accelerator Supplement: RGPAS-2020-00031), Healthy Brains, Healthy Lives (New Investigator Award: 2b-NISU-8) (B.A.R.); CIFAR Learning in Machines and Brains Program (K.K. & B.A.R.); Canada CIFAR AI Chair program (G.L. & B.A.R.). This work was enabled by the material support of NVIDIA in the form of computational resources and support provided by Mila.

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
