# OpenReview forum: "How gradient estimator variance and bias impact learning in neural networks"
_ICLR.cc/2023/Conference — ICLR 2023 poster_

### Official Review · Reviewer_27vS · 2022-10-25

**Confidence:** 4
**Correctness:** 3
**Technical Novelty And Significance:** 3
**Empirical Novelty And Significance:** 3
**Recommendation:** 8

**Clarity, Quality, Novelty And Reproducibility:**

The paper is clear, except for the used mathematical notations, which need to be improved. The experiments are reproducible since the details are mentioned in the appendix. The degree of novelty of this work is unclear since the authors didn’t address most of the related works (e.g., perturbed gradient descent and stochastic gradient Langevin dynamics) or the larger history of research on adding noise to improve learning (e.g., Murray and Edwards 1992). This paper needs to be well-distinguished from other works, which can happen by carefully assigning credit to the earlier papers in addition to mentioning how your paper’s contributions are different from earlier contributions.

**Strength And Weaknesses:**

**Strengths**:
1. The problem tackled by this paper is fundamental and important. The paper shows a promising direction that can lead to more impactful works.
2. The paper provides a principled way of analyzing how bias and variance impact learning using a careful setup based on the change in loss with an update having intrinsic known bias and variance compared to the change of the loss using the update based on true gradient.
3. The paper has substantial results explaining many of the standard design practices in deep learning in a more grounded way.
4. The paper is well-written, and the problem is well-motivated.

**Weaknesses**:
1. Unclear notations. The authors used the same notations to write vectors and scalars. Reading these notations would be challenging to follow for many readers. Please consider updating your notations and refer to the notation section in the Formatting Instructions template for ICLR 23.
2. The framework impact is unclear. The authors mentioned that the case of intrinsic but known bias and variance is often the case in computational neuroscience and neuromorphic engineering. This is the main motivation for their approach. However, the framework provided is limited to specific cases, namely, white noise and fixed bias.
The authors argue that their assumptions are reasonable for most cases computational scientists and neuromorphic engineers face, but they don’t provide evidence for their claims. Clearly, this framework provides an important way for analyzing methods such as perturbed gradient descent methods with Gaussian noise, but it’s unclear how it can help analyze other cases. This suggests that the framework is quite limited.
	* The authors need to show that their choices and assumption are still useful for computational neuroscience and neuromorphic engineering. This can happen by referring to contributing and important works from these fields having known bias and variance with Gaussian noise.
	* In the experiments, the used bias is restricted to having the same magnitude for all weights ($b \vec{1}$). Can we reproduce the results if we use arbitrary biases? It would be better if the authors tried a number of arbitrary biases and averaged the results.
3. The paper is not well-placed in the literature. The authors didn’t describe the related works fully (e.g., stochastic gradient Langevin dynamics). This makes the work novelty unclear since the authors didn’t mention how analyzing the gradient estimator was done in earlier works and how their contribution is discernible from the earlier contributions. Mentioning earlier contributions increases the quality of your work and makes it distinguishable from other work. Please also refer to my comment in the novelty section.
4. Missing evidence of some claims and missing details. Here, I mention a few:
	* It’s not clear how increasing the width and/or depth can lower the trace of the Hessian (Section 2.1). If this comes from a known result/theory, please mention it. Otherwise, please show how it lowers the trace.
	* The authors mentioned that they use an analytical and empirical framework that is agnostic to the actual learning rule. However, the framework is built on top of a specific learning rule. It’s unclear what is meant by agnostic in this context (Section 1).
	* The authors mentioned in the abstract that the ideal amount of variance depends on the size and sparsity of the network, the norm of the gradient, and the curvature of the loss landscape. However, the authors didn’t mention the sparsity dependence anywhere in the paper.
	* The authors mentioned in a note after the proof of Theorem A.5 that it is also valid for Tanh but not Sigmoid. However, the proof assumed that the second derivative is zero. It’s unclear whether a similar derivation can be developed without this assumption. However, the authors only mention the relationship with the gain of $\phi(.)$.
	* More information is needed on how the empirical likelihood of descent is computed (Fig. 7).
	* The use of MSE should be mentioned in Theorem A.3 since it’s not proven for any loss function. In addition, the dataset notation is wrong. It should be $\mathcal{D}=\\{(x_1, y_1), ..., (x_M, y_M)\\}$, where $M$ is the number of examples since it’s a set containing input-output pairs, not just a single pair.
	* The argument in Section 2.1 that increasing depth could theoretically make the loss less smooth is not related to the argument being made about variance. It is unclear how this is related to the analyses of how increasing depth affects the impact of the variance. I think it needs to be moved in the discussion on generalization instead.
5. A misplaced experiment that does not provide convincing evidence to support the theorems and lemmas developed in the paper with less experimental rigor (Fig. 1).
	* The experiment is misplaced being at the introduction section. This hurts the introduction and makes the reader less focused on your logic to motivate your work.
	* It’s not clear from the figure what the experiment is. The reader has to read appendix B2 to be able to continue reading your introduction, which is unnecessary.
	* The results are shown with only three seeds. This is not enough and cannot create any statistical significance in your experiment. I suggest increasing the number of runs to 20 or 30.
	* It’s unclear why batch gradient descent is used instead of gradient descent with varying bias and variance. Using batch gradient descent might undesirably add to the bias and variance.
	* The experiment results are not consistent with the rest of the paper. We cannot see the relationship when varying the bias or variance similar to other experiments. Looking at Fig.1B where bias=0, for example, we find that adding a small amount of variance reduces performance, but adding more improves performance up to a limit. This is not the case with the other experiments, though. I suggest following the previous two points to make the results aligned with the rest of your results.
6. Alternative hypotheses can be made with some experiments. The experiment in Fig. 3.A  needs improvement. The authors mention that excessive amounts of variance and/or bias can hinder learning performance. In Fig. 3. A, they only show levels of variance that help decrease loss. An alternative explanation from their figure is that by increasing the variance, the performance improves. This is not the case, of course, so I think the authors need to add more variance curves that hinder performance to avoid alternative interpretations.

**Minor issues that didn’t impact the score**:
* There are nine arXiv references. If they are published, please add this information instead of citing arXiv.
* What is a norm $\sqrt{N}$ vector? Can you please add the definition to the paper?
* You mentioned that the step size has to be very small. However, in Fig. 1, the step size used is large (0.02). Can you please explain why? Can this be an additional reason why there is no smooth relationship between the values of the variance and performance?
* No error bars are added in Fig. 4 or Fig. 7. Can you please add them?
* In experiments shown in Fig. 3 and Fig. 5, the number of runs used to create the error bars is not mentioned in Appendix B.2.
* A missing $\frac{2}{|\mathcal{D}|}$ in Eq. 27.
* In Theorem A.3 proof, how the input $\mathbf{x}$ has two indices? The input is a vector, not a matrix. Moreover, shouldn’t $\sum_k (W_k^{(2)})^2 = 1/d$, not $d$?


**Summary Of The Paper:**

Many methods rely on approximations of the true gradient, which have intrinsic bias and variance. This paper shows how bias and variance of gradient estimators can hinder or improve learning. Although bias and variance can potentially impair learning, the right amount can be helpful for learning and for generalization. The effect of bias and variance seems to be independent of each other, so this paper studies each separately. The paper provides a framework for studying the effect of adding a fixed bias and white noise in the update rule. Given the bias and variance of a learning algorithm, one can use these guidelines to find the network design that facilitates learning.

**Summary Of The Review:**

The work is well-written and well-motivated. The paper tackles a fundamental problem and provides a principled way of analyzing gradient estimators. The proposed framework has substantial results explaining many of the best/standard design practices in deep learning in a grounded way. However, the paper is lacking some mathematical rigor when it comes to notations. The proposed framework has an unclear impact on giving guidelines for the targeted fields, namely computational neuroscience, and neuromorphic engineering, and some arguments are needed to justify how it serves the intended purpose. More arguments and examples are needed to justify how the assumptions are not limiting. In addition, the paper is not well-placed in the literature, which makes the novelty of the work unclear. More careful work is needed to distinguish this work from others. There is also some missing evidence for some claims made by the authors and some imprecisions. Finally, the paper has some experimental inconsistencies that contradict other results but can be fixed easily.

That being said, I think substantially improving the quality of this paper seems well possible by addressing this feedback. All my comments can be addressed by the authors since they are not affecting the main arguments of the paper. This paper has potential, and I would be willing to change my scores, subject to the authors’ clarifications and incorporating the requested changes.

---

> ### Author Response · Authors · 2022-11-17
> **Response to Reviewer 27vS 1/4**
>
> We thank the reviewer for accurately summarizing our work and appreciating the principled nature of our approach. We would also like to thank the reviewer for their questions and concerns and below, we respond to each comment in detail:
>
> 1. **Unclear notations:** We thank the reviewer for making this point, and see what they mean. We will update the notation to make vectors and scalars different per the ICLR 2023 guidelines.
>
> 2. **Potential limitations of the framework:** The reviewer is correct that we made some assumptions about the noise and bias terms in order to permit our mathematical analysis. In carefully examining the reviewer’s interpretation of our assumptions, we agree that we should make the generality of these assumptions clearer.
>
>     - First, with respect to the noise, the only real assumptions we made were that the noise is zero in expectation and iid across parameters (which hold for Gaussian noise, of course). It should be noted that there may be learning algorithms that do not adhere to these assumptions, which does mean we cannot state that our analysis is completely learning rule agnostic. Nonetheless, these assumptions (zero in expectation and iid) are sufficiently general that they apply to a number of existing algorithms (e.g. noise perturbation, AGREL, and EQ-prop with the beta parameters selected to have expectation of zero), and we suspect they can apply to many more. As per the reviewer’s astute comment, we will add some additional clarification on these limitations, and will point out which algorithms our assumptions do hold for, in the Appendix.
>
>     - Second, with regards to the bias, we can see that we were insufficiently clear about how our  theorems extend to an arbitrary bias. Our proofs all assume an arbitrary bias vector, and they do not assume the bias is constant because the proofs focus on a one-step expansion. In fact, our proofs still hold for algorithms where the bias vector is different at each update step, as assumed in the multi-step analysis (see Corollary A.2).
>
> 3. **The paper is not well-placed in the literature:** We thank the reviewer for pointing this out and suggesting how we can be more thorough about the relationship between our work and previous work on the dynamics of learning. In our initial submission, we tried to do some of this, as seen in the Discussion and Appendix C section we highlight the relationship to previous work on SGD dynamics. However, we forgot to include any mention of earlier work on Langevin dynamics, as the reviewer points out. We thank the reviewer for highlighting this gap, and we will now include such material in the Appendix.

---

> > ### Author Response · Authors · 2022-11-17
> > **Response to Reviewer 27vS 2/4**
> >
> > 4. **Missing evidence of some claims and missing details:**
> >
> >     - Relationship between trace of Hessian and network width/depth: We thank the reviewer for bringing this point up and we hope our updated manuscript resolves this by better clarifying that a *smoother loss landscape is linked to a lower trace of the loss Hessian* (see Yao et al. 2018, cited in the paper). In light of this result, we described in Section 2.1 (2nd paragraph) that it has been empirically shown in existing studies that increasing width leads to smoother loss landscapes, thereby implying a lower trace of loss Hessian. For depth, pertinent results can be found in Theorem 2.2 and A.4. We would like to emphasize that **we are examining the effect on $\frac{Tr(Hessian)}{N}$ in Theorem 2.2 instead of just $Tr(Hessian)$** (notice the N in the denominator). Thus, although some existing studies (e.g. Li et al., 2018) indicated that increasing the depth could make the loss landscape less smooth, and thereby could lead to increased trace of loss Hessian, we show in Theorem 2.2 how increasing depth leads to reduced impact of variance because both numerator, $Tr(Hessian)$, and denominator, $N$, increase when increasing depth of a network. We will update the manuscript accordingly.
> >
> >     -  The framework is built on top of a specific learning rule: Respectfully, we don’t see how the reviewer came to this conclusion. It is true that we make some assumptions about the learning rule, namely that it is additive (as opposed to multiplicative), and that its noise has zero mean and is iid (as discussed above). The reviewer is correct that this prevents the analysis from being 100% learning rule agnostic, and we were arguably insufficiently clear about this point. We will clarify this in Section 2. We thank the reviewer for drawing our attention to this, as it is important to be clear to readers regarding this fact. But, we wish to reiterate that nothing in our analysis assumes a *specific* learning rule, rather, it assumes a broad class of learning rules.
> >
> >     - Dependence of impact of variance on network sparsity: This comment suggests that we were not sufficiently clear as to what we meant by sparsity. For that reason, we would like to clarify that sparsity here is in terms of activity instead of connectivity; we will clarify that in the manuscript. For activity-based sparsity,  we do in fact show, in Theorem 2.3 that a network where some of the units have zero activity will have less of an impact from variance, and we take this to be part of a sparse representation, i.e., you cannot have a sparse representation without some units taking a value of zero in their activity. But, this interpretation was not laid out clearly in Theorem 2.3, and we thank the reviewer for highlighting that fact and helping to improve the manuscript in this regard.
> >
> >     - Theorem A.5 for tanh non-linearity: We thank the reviewer for pointing this out. We realized that the second derivative of tanh cannot be ignored and appears as a factor in the Hessian. Moreover, the term containing the second derivative of tanh would also consist of the gradient of the loss function, thereby making it very difficult to infer the relationship of the Hessian traces with and without the tanh non-linearity. Consequently, we will now remove the sentence about tanh from the note in the Appendix. We would like to point out that this change does not affect our core claims around network activity sparsity (because tanh does not sparsify activations). Nevertheless, we sincerely thank the reviewer for catching this discrepancy in the claim about tanh and the derivations.
> >
> >     - More information on how empirical likelihood of descent is computed (Fig. 7): We thank the reviewer for this suggestion, we will increase the exposition on this matter in the appendix.
> >
> >     - The use of MSE should be mentioned in Theorem A.3 since it’s not proven for any loss function. In addition, the dataset notation is wrong: The reviewer is correct, we should mention the use of MSE in Theorem A.3, and we will add it. As well, we will change the dataset notation per the reviewer’s recommendation. We thank the reviewer for these excellent suggestions.
> >
> >     - Argument that increasing depth could theoretically make the loss less smooth: We can see now that we were unclear in how we wrote this section. The relationship between the effect of network depth on loss landscape smoothness and the impact of variance is mediated by the trace of the Hessian, namely, a smoother loss landscape is associated with a lower trace of the Hessian, which in turn implies a lower impact of the variance (Theorem 2.2). Our point here was simply that increasing the depth presents a complication in this respect, as it may increase the trace of the Hessian. We will clarify that in Section 2.1.

---

> > > ### Author Response · Authors · 2022-11-17
> > > **Response to Reviewer 27vS 3/4**
> > >
> > > 5. **Misplaced experiment without convincing evidence to support the theorems and lemmas developed in the paper (Fig. 1):** Thanks to the reviewer’s comment, we realized that our point with this first figure is insufficiently clear. Our point with this first figure was not to verify our theorems, per se. We would also like to point out that we are using full-batch GD (different from the standard mini-batch GD used in standard deep learning training protocols), wherein we compute the gradient over the entire dataset before making an update to the network parameters. Therefore, we believe that our experiment protocol is similar to what the reviewer has in mind, and the confusion about “batch GD” is probably an issue with terminology, i.e. full-batch GD is what we call non-stochastic GD.  In this figure, our point was merely to demonstrate two basic facts that many researchers already know, but which motivated our analyses, namely, that too much bias and variance harms training, but a small amount can aid it. We think that this is an important point to make at the beginning of the paper for any readers who are not aware of these facts, though. Nonetheless, we see the reviewer’s point about the small number of seeds. To that end, we have now re-created this figure with 20 seeds, and the same results hold, i.e. too much bias/variance impairs learning but a little bit can help. Of course, even with 20 seeds there is still some stochasticity, such that the impact of adding bias or variance is not purely smooth (more on this below). In an ideal world, we would be able to run enough seeds to cut through this noise, but we do not have the compute resources nor time available to run more than 20 seeds before the rebuttal deadline. Yet, we believe that this improved figure with more seeds, along with some clarifying sentences that we will add to the introduction, greatly improve the utility of this figure. If the reviewer disagrees, we can always move the figure to the Appendix, but we do feel that it is important to clarify this point for some readers who may not have considered it before.
> > >
> > > 6. **Fig. 3A only shows levels of variance that help decrease loss:** We believe that the issue here is actually a lack of clarity on our part. Figure 3A is plotting the Delta Loss, and Delta Loss does not measure the decrease in loss, rather it measures the difference between the loss decrease that one obtains with true gradient descent and the loss one obtains with a given level of bias and variance, on the training data (see Equation 2). Thus, Fig. 3A does in fact show variance hindering learning performance, not aiding it, since a positive Delta Loss implies *less* of a drop in loss. We will clarify this in the legend for Fig. 3A, and add the arguments and temporal sub-script to all mentions of Delta Loss in the paper, in order to make clear that this value refers to the value defined in equation 2, not just the change in loss. We thank the reviewer for highlighting this issue.

---

> > > > ### Author Response · Authors · 2022-11-17
> > > > **Response to Reviewer 27vS 4/4**
> > > >
> > > > 7. **Small points:**
> > > >
> > > >     - Updating the arXiv references: We thank the reviewer for pointing this out and have now updated the articles with their published versions.
> > > >
> > > >     - What is a norm $\sqrt{N}$  vector: It refers to a vector whose 2-norm is equal to the square root of $N$. We will add this clarification in the Appendix.
> > > >
> > > >     - Lack of smoothness in Fig. 1: We thank the reviewer for suggesting possible explanations for the lack of smoothness in Fig. 1 plots. We ran further experiments and believe that we have a better understanding of this phenomenon. Firstly, a learning rate of 0.02 was used to achieve reasonable performance in available compute time (training epochs) while training the VGG-16 network on CIFAR-10. Using a lower learning rate requires a significantly longer training protocol to demonstrate that adding excess noise or bias to the gradient is severely detrimental to performance. Furthermore, we would like to point out (as already mentioned before) that the purpose of Fig. 1 is to demonstrate that the phenomenon we are studying is relevant even in practical scenarios and it is common to use learning rates of similar order along with a gradient descent optimizer. Secondly, we would expect Fig. 1 to be smooth only if the loss landscape is extremely well-behaved, such that progressively adding some noise or bias helps avoid a minima and converge to a nearby minima with slightly different performance than the previous one. We believe that this behavior, although attractive while reasoning, is extremely unrealistic in practice, more so when the experiment is repeated with different initializations. Finally, we believe that this non-smooth and seemingly complicated behavior demonstrated in Fig. 1 sets up the tone of the manuscript and conveys to the reader that it is indeed a non-trivial problem and provides a foray into our theorems and results that explain this effect in greater detail.
> > > >
> > > >     - No error bars are added in Fig. 4 or Fig. 7: Fig 4 is actually a scatter plot with all points that were empirically obtained. We believe that a scatter plot is more informative in this case instead of a line plot of means with error bars. Fig 7 indicates the empirical probability of descent, i.e. it indicates the number of times we found a decrease in the loss divided by the total number of times an update was done, under the biased approximation of the gradient. We are not sure what an error bar in this context would indicate.
> > > >
> > > >     - Number of runs used for error bars in Fig. 3 and Fig. 5: For all experiments involving the student-teacher toy dataset setting, i.e. Fig. 3 and Fig. 4, we ran at least 20 different seeds for creating the dataset and 20 different seeds for network initialization. Therefore, we ran at least 400 different runs to generate the figures. For the experiments involving VGG networks and CIFAR dataset, we ran at least 10 seeds determining the network initialization for each experiment (mentioned in the appendix). The number of seeds was chosen given the experiment feasibility in available compute resources and time. We will include these details in Appendix B.2.
> > > >
> > > >     - A missing $\frac{2}{|D|}$ in Eq. 27: We sincerely thank the reviewer for their astute comment. However, the factor $\frac{2}{|D|}$ is included in the expression for trace of 1-layer network (see Eq. 26).
> > > >
> > > >     - Clarifications around Theorem A.3 proof: We would like to again apologize for the confusion around notation of vectors, matrices and scalars. Here, $x_{nk}$ denotes the $k^{th}$ dimension of the $n^{th}$ input. For the second question, could the reviewer kindly point to which line they are referring to? As a clarification, we have used $\sum_k (W_k^{(2)})^2 \approx 1$ in our proof (last paragraph on Page 18).
> > > >
> > > > We sincerely thank the reviewer again for their helpful comments and suggestions that have helped us improve the overall quality of the manuscript. We hope that the reviewer will find our clarifications helpful and rate our work to be of acceptance quality at ICLR.

---

> > > > > ### Comment · Reviewer_27vS · 2022-11-26
> > > > > **Thank you for your response**
> > > > >
> > > > > Thank you for the detailed response and for addressing the issues I mentioned in my review. The authors clarified some details and corrected the mistakes. They addressed my concerns and updated their manuscript to reflect that. Therefore, I changed my score, and I would like to see this paper published.

---

> > > > > > ### Author Response · Authors · 2022-12-11
> > > > > > **Thank you**
> > > > > >
> > > > > > We are glad to know that you found our response and changes to the manuscript to be helpful. We would like to thank you again for your detailed and helpful review. We strongly believe that your comments have improved the quality of the manuscript and are excited to see that you believe the same.

---

### Official Review · Reviewer_HbzZ · 2022-10-25

**Confidence:** 3
**Correctness:** 3
**Technical Novelty And Significance:** 2
**Empirical Novelty And Significance:** 3
**Recommendation:** 5

**Clarity, Quality, Novelty And Reproducibility:**

Clarity & Quality: There is good clarity in terms of mathematical equations for examining loss, variance and bias and how it affects learning . These are well explained followed up experimental backing and data analysis to support the proofs/theorems for impact on training, loss and across different ranges of networks and generalization. Clear arguments of why proxy metric L is necessary which is differentiable given fairness constraints are non-convex

Novelty: From a novelty perspective this paper does not have much to offer as existing concepts such as bias variance, loss and their impact is revisited across different learning paradigms, explained rigorously with math proof.

Reproducibility: There isn't a section made available on how to reproduce the experimental setup to quantify measurements and validate the results proposed in the paper.

**Strength And Weaknesses:**

Strengths:
1. Mathematic basis and experimental setup is solid, supported by strong analysis  to demonstrate impact of variance on wider deeper networks which is low
2. Sound empirical verification of proposed theorems to assess the impact of bias/variance  generalization.
3. Great summarization of relevant work


Weakness:
1. Empirical evaluation done only on artificial neural networks, could be extended to more architectures.
2. Does not have any in depth analysis of any biologically focussed learning algorithms (acknowledged by author)
3. While it acknowledges the impact of bias/noise on learning performance, it does not provide any plans to eliminate noise/variance
4. There isnt any plan to improve learning algorithms as well in the advent of bias/noise impact on them (acknowledged by author)
5. There is an assumption that Loss L is twice differentiable but no justification for this assumption.


**Summary Of The Paper:**

The authors in this paper discuss the pros and cons of how variance and bias affect gradient estimator, to better understand how gradients are approximated by brains. The authors argue that while bias and variance generally affect training data learning negatively, some amount of bias, variance can be beneficial for generalization. The authors try to quantify bias/variance in relation to several factors such as size and sparsity of  the network, gradient norm and loss landscape curvature. The authors feel some of these factors can constructively help guide training of biologically inspired networks or neuro chips, especially for designing gradient estimators. The authors propose that bias & variance prevent a network from converging to a sharp minima. This leads to better generalization. THey also show that the imapct of bias has a linear relationship with the norm of the gradient of the loss function, and is influenced by the gradient direction and the loss Hessian's eigen vectors.
The authors also show that the impact of variance is to a lesser degree and low for those networks with a threshold non-linearity. They also show that variance is least influential for deep/narrow networks.



**Summary Of The Review:**

My recommendation is to not accept the paper, unless a few revisions are made which are highlighted in the weakness section. This paper illustrates the importance of the impact of variance and bias on gradient estimators and how it impacts learning. While it definitely improves the understanding by proposing mathematical theorems and providing explanations via analysis, it is not clear to me how it is novel and advances the field and how bias or noise and their detrimental impact can be reduced. I think if some of these problems are addressed via additional work, it could be a great paper.

---

> ### Author Response · Authors · 2022-11-17
> **Response to reviewer HbzZ 1/2**
>
> We thank the reviewer for their comments and for appreciating the mathematical rigor and empirical setup of our paper.
>
> We would like to point out one small correction in the reviewer’s summary of our work, which we believe could help to clarify the scope of the work and address some of the concerns raised by the reviewer. In their summary, the reviewer concludes that we quantify the bias/variance in relation to several design choices and that our work could help to guide researchers that are designing gradient estimators. However, in fact, we are neither trying to quantify the amount of noise or bias, nor how it depends on the network properties. As well, we are not suggesting guidelines for designing new learning rules. Instead, we are trying to determine how network design choices affect the impact of noise and bias on training. We are doing this in order to provide guidance to researchers *who already have a gradient estimator of choice*. By using our findings, they can make design choices in their networks in order to minimize the negative impact of any noise and bias in their gradient estimator, while maintaining some of the potential positive effects.
>
> In light of this clarification, we would like to note that many of the weaknesses identified by the reviewer were explicitly highlighted in our limitations section as things that may be desirable, but which are outside the scope of the current work. Given that no paper can do everything, we feel that proper scientific rigor demands just this, i.e. clarity on limitations. Nonetheless, we respond to the comments in detail below and outline the corresponding changes to the manuscript that have been, or could be, made.
>
> 1. **Empirical evaluation done on artificial neural networks:** To be candid, we are not sure what the reviewer means by this comment, so we would appreciate it if the reviewer could please let us know if in our response we misunderstood. It is true that we only test on artificial neural networks (ANNs), but we are not sure what the alternatives are that the reviewer is imagining, given that we cannot test these ideas in actual biological tissue, nor do we have neuromorphic chips at our disposal. To be clear, we included data from both fully connected neural networks (Figs. 3, 4, and 7) and convolutional neural networks (Figs. 1, 5, and 6) of varying sizes, so we were not testing our ideas on a single architecture. That said, perhaps the reviewer had in mind other potential architectures, such as recurrent neural networks (RNNs)? As we noted in the Discussion, restricting the empirical results to feedforward networks is indeed a limitation in the manuscript. With the limited time available for responding to reviewer comments in ICLR we cannot produce a great deal of new data, but, if the reviewer feels it is important we can verify that one of our core findings replicates in RNNs, namely, that the impact of variance is reduced in larger networks. We can include this new figure in the supplementary material of the revised manuscript, if the reviewer requests it.
>
> 2. **Lack of analysis or improvements to biologically focussed learning algorithms:** We appreciate the reviewer for pointing out this limitation of our work and are happy that they share our desire to use the proposed framework to analyze biologically-plausible learning rules. But, in this paper, our goal as stated was to provide a learning algorithm agnostic analysis (see Introduction, page 2). In contrast, an analysis of a specific biologically focussed learning algorithm (or some class of such algorithms) would prevent a learning rule-agnostic approach. In a much longer manuscript (longer than ICLR papers), it may be desirable to follow-up our rule-agnostic analysis with a case study on a specific algorithm. However, that is outside the scope of the present manuscript. Nonetheless, we believe that our work will provide the basis for researchers interested in reducing the impact of bias and variance on their networks, e.g. by guiding them in their selection of network design (width, depth, activation function, etc.). Thus, though we share the reviewer’s instinct that analysis of a specific learning rule would be desirable we respectfully do not consider this to be within the appropriate scope of this paper.

---

> > ### Author Response · Authors · 2022-11-17
> > **Response to reviewer HbzZ 2/2**
> >
> > 3. **Plans to eliminate noise and bias:** We thank the reviewer for bringing up this point which will help us to clarify the scope of our contributions. The response to this comment is similar to the response to comment 2. Although we appreciate the reviewer’s suggestion and are personally excited by potential research into how to reduce bias and variance in biologically plausible or neuromorphic learning algorithms, that was not the purpose of this paper. The purpose of this paper is instead to provide guidance for researchers who already have a learning algorithm that they are using as to the design choices they can make in their networks to minimize the impact of any bias and variance in their learning algorithm. Put another way, our current work is not targeted at designing new learning algorithms, but rather, designing networks to work best with existing learning algorithms. Taken together, our contributions can be viewed as principles that can guide decisions about network design (depth, width, activation non-linearity) if a learning rule is known to have some non-trivial degree of noise and bias in its gradient estimator.
> >
> > 4. **Novelty of results:** We appreciate the concerns raised by the reviewer around the novelty of our work and, respectfully, we disagree with them. Past work on this topic, that we are aware of, has all been done on a learning-rule specific case-by-case basis (e.g. the paper by [Werfel *et al.*, 2003](https://proceedings.neurips.cc/paper/2003/file/f8b932c70d0b2e6bf071729a4fa68dfc-Paper.pdf)). As well, this past work typically only considered the variance of unbiased algorithms. In contrast, our paper operates in a learning rule agnostic manner and considers the impact of both variance and bias, both of which are important to biologically plausible and neuromorphic learning algorithms (see Introduction 2nd paragraph for more details on differences between our work and the deep learning theory or optimization literature). This is, to our knowledge, a completely novel contribution. Nonetheless, given the reviewer’s comment, we may not have made sufficiently clear this relationship to other work. Therefore, we will add some text to the discussion to clarify this relationship and make the novelty of our work more explicit for readers.
> >
> > 5. **Reproducibility:** We agree with the reviewer that reproducibility is critical and we thank the reviewer for this comment. To this end we provide thorough experimental details in the appendix, and in the supplementary materials we have code to run all of the experiments and generate the figures. We believe that this information will allow the community to validate/replicate our results and further build on our work.
> >
> > 6. **Assumption that L is twice differentiable:** We thank the reviewer for their astute observation. Our reasons for having this assumption are two-fold. Firstly, we believe that it is a standard assumption in many theoretical analyses of deep learning and optimization frameworks, such as Raman *et al.*, 2019. We would also like to note that common loss functions used for training deep networks, e.g. mean squared error or cross entropy, are indeed twice differentiable. Secondly, this assumption allows us to apply a second order Taylor series expansion, which in turn forms the basis of all our derivations. We agree with the reviewer that it is indeed worth asking how our results would transfer if the loss is not twice differentiable. However, we believe that this question falls outside the scope of our work and does not restrict our work from being useful to computational neuroscience and neuromorphic researchers alike, because most of the common loss functions are covered by our analysis.
> >
> > We thank the reviewer again for their helpful comments and suggestions. We hope that the reviewer will find our clarifications helpful in contextualizing our contributions and rate our work to be of acceptance quality at ICLR.

---

> ### Author Response · Authors · 2022-12-11
> **Follow-up on response**
>
> Dear Reviewer HbzZ,
>
> As the discussion period is ending soon, we were wondering if you had any further questions or comments to which we can respond. We would be interested to know if our response addressed your concerns about our manuscript, specifically if our response clarified the questions you had about our work. If so, we would be extremely grateful if you could consider revising your score to reflect the same.
>
> Thank you,
> Authors

---

### Official Review · Reviewer_Yx5o · 2022-10-26

**Confidence:** 2
**Correctness:** 3
**Technical Novelty And Significance:** 3
**Empirical Novelty And Significance:** 3
**Recommendation:** 8

**Clarity, Quality, Novelty And Reproducibility:**

Clarity: The paper is well-written and was easy to read.
Quality & Novelty: While some related analysis may exist in related literature, the confirmation of the theory with experiments makes an overall compelling study.
Reproducibility: It is not clear if the results are reproducible and it seems important for the code to be released.

Minor comment:
- \hat n is a unit-length vector? unit-vectors are of referred to as basis elements


**Strength And Weaknesses:**

Strengths:
---
* The theoretical results seem interesting and, importantly, are validated in a series of experiments.
* The proposed theory may be useful to find neural architectures which work well for a given learning algorithm.

Weaknesses:
---
* Figure 1 was not very illuminating except for the sharp phase transition occuring when bias and/or variance get too large. As a red/green-blind person, it was very difficult to tell apart the areas which improve upon GD and it looked more or less random. It is important that these experiments are conducted over many different random seeds and then averaged, to make sure the differences statistically significant. Perhaps the image in Fig. 1 will look less "noisy" if more random seeds are considered? If the computational resources are available, it would be great to have a higher-resolution plot.



**Summary Of The Paper:**

The paper studies how bias and variance of gradient noise can impact the generalization performance. Specifically, the paper proposes a theory which connects bias and variance to network size, activation functions, gradient norm and Hessian. This theory is shown to coincide with empirical results.

**Summary Of The Review:**

The paper provides an interesting and useful analysis of noise and variance of learning algorithms. The theory is shown to match practice on some learning problems.  I believe the paper makes an interesting and useful contribution and I therefore recommend acceptance.

---

> ### Author Response · Authors · 2022-11-17
> **Response to Reviewer Yx5o**
>
> We thank the reviewer for their encouraging feedback of our work and appreciating the core contributions of our work. We are also grateful to the reviewer for pointing out avenues for improvement and we propose to address these issues in the updated version of our manuscript. Below we respond to the reviewer’s concerns:
>
> 1. **Figure 1:** We sincerely apologize for the color scheme used in Fig. 1 and the difficulty it posed to the reviewer. We will update the figure with a different color scheme and hope it is easier to read. Owing to limited computational resources and time, we were restricted to run only 3 seeds per configuration (noise and bias levels). We have now added more seeds to our analysis (20 seeds in total) to increase the robustness of our empirical observations. Although the results are still noisy, we hope it is more convincing to the reviewer. We believe that Fig. 1 would have shown a smooth relationship between varying amount of noise and bias if the loss landscape is extremely well-behaved, such that progressively adding some noise or bias helps avoid a minima and converge to a nearby minima with slightly different performance than the previous one. We believe that this behavior, although attractive while reasoning, is extremely unrealistic in practice, more so when the experiment is repeated with different initializations. Through this Figure, we want to indicate to the reader that some amount of approximation errors might be beneficial to performance although too much is detrimental. We believe that this figure also indicates that figuring out the amount of noise and/or bias that is beneficial is a non-trivial problem and provides a foray into our theorems and results that explain the impact of approximation error in greater detail.
>
> 2. **Reproducibility:** Our appendix contains specific sections corresponding to experiment details that outline the details underlying each figure. We have also included the code in the supplementary materials and believe that it serves as a good resource for the community to reproduce our results.
>
> 3. **Notations:** Yes, $\hat{n}$ indicates a unit-length vector. We apologize for the confusion between unit-length and unit (basis) vectors. In this context, we wanted to indicate the direction of the noise vector with a unit-length vector, which does not need to be aligned to one of the cardinal axes or basis vectors.
>
> We thank the reviewer again for their helpful comments and suggestions for improving our manuscript. We hope that the reviewer will find the updated version of the manuscript to be ready for acceptance to ICLR.

---

> > ### Author Response · Authors · 2022-11-18
> > **Updated Figure 1**
> >
> > We would like to once again apologize for any inconvenience the reviewer faced due to the colorscheme used in Figure 1. We have now added a version of Figure 1 in Appendix B.1 that uses the Spectral colorscheme. We hope that these two figures, taken together, will improve the readability of the Figure and will help us get the main takeaway (as decribed in the previous comment) across to the reader. We hope the reviewer finds the new version of the manuscript more reader-friendly.

---

### Official Review · Reviewer_Z5BT · 2022-10-26

**Confidence:** 3
**Correctness:** 3
**Technical Novelty And Significance:** 3
**Empirical Novelty And Significance:** 2
**Recommendation:** 6

**Clarity, Quality, Novelty And Reproducibility:**

The paper is generally well-written and easy to follow. The authors provide novel and strong results, they further validate these results by empirical experiments. Unfortunately, some missing information regarding the experiments (as mentioned in (a1) under weaknesses above) makes the results irreproducible.

**Strength And Weaknesses:**

Strengths:
(a) The authors study an interesting and generic problem, which will have wide influence on the community, given that training neural networks via stochastic gradient descent methods are at the center of attention in machine learning.
(b) The authors present strong and insightful theoretical results.
(c) The authors validate their results via numerical experiments.

Weaknesses:
(a) I had difficulty understanding and interpreting some key parts in the paper, in particular:
      (a1) Early on the authors define delta L in Eq. (2) as difference between loss changes when performing the weights with bias and variance (via E1. (1)) versus when the weights are updated using the full gradient. It is not clear to me whether the delta L reported in Figures 3 to 6 is this quantity or simply the changes on the loss function at each step. If the former is the case, did you update the weights via Eq. (1) with and without bias and variance and then computed the difference in loss functions at every step?
     (a2)  In Figures 5 and 6 I was not able to find which bias (b) and variance (sigma) these results correspond to.
     (a3) In Sec. 3.1, the authors claim that updating weights by adding bias and variance could lead to "flatter" points w.r.t. the loss function and hence better generalization. In particular, in Theorems 3.1 and 3.2 they provide some conditions for the decrease in loss functions and then they show both theoretically and empirically that by adding bias and variance the decrease in loss function is less likely. However, I am not convinced that this is sufficient for showing that adding bias and variance would lead to better generalization. Perhaps reporting some metrics or the loss value for test sets would strengthen these claims.





**Summary Of The Paper:**

The authors study effects of bias and variance of gradient estimators for training neural networks that is typically done via gradient descent and its variants. In particular, the authors provide novel and intersting theoretical results on the effects of gradient estimators bias and variance on changes in loss function. They further confirm these results by numerical experiments.

**Summary Of The Review:**

Overall, the paper presents novel and strong results on the effects of bias and variance of gradient estimators on training deep networks via gradient methods. The main weaknesses of this paper are (a) missing details as explained under the weaknesses section) (b) slightly overstating the importnace of bias and variance in better generalization results.

---

> ### Author Response · Authors · 2022-11-17
> **Response to Reviewer Z5BT**
>
> We thank the reviewer for their encouraging feedback of our work and highlighting the core strengths of our submission. We are also grateful to the reviewer for pointing out avenues for improvement and we propose to address these issues in the updated version of our manuscript. We believe that the reviewer’s rating of our work primarily reflects their concerns around the presentation clarity. Therefore, we respond in detail to the specific concerns raised by the reviewer and hope that it convinces them to increase their rating:
>
> 1. **Definition of $\Delta L$ in text and figures:** The plots follow the same definition of $\Delta L$ as described in text (equation 2). As pointed out by the reviewer, we empirically measured $\Delta L$ by computing the difference in loss with and without the noise/bias term. We have described the experimental details in the Appendix and added the code as supplementary material. But, we can also see how it might be natural to think that $\Delta L$ refers to the change in loss. Thus, we will update the text and figures to add the time-step subscript and bias/variance arguments for $\Delta L$, so as to make it more clear that this is the same quantity as indicated by equation 2.
>
> 2. **Bias and variance values in Fig 5 and 6:** We apologize for this lack of detail in the Figures and will add the relevant details in the figures and the text. For completeness, the variance value used in Fig. 5 is 20 and the bias value used in Fig. 6 is 0.02. We will also add this information under experimental details for each figure in the appendix, which we believe will help in improving the overall reproducibility of our work.
>
> 3. **Bias and variance leading to better generalization:** We thank the reviewer for their helpful suggestion in connecting our theoretical results with previously mentioned empirical findings. Specifically, Fig 1 shows that there can be some gains in performance when moderate amounts of noise and bias are added to the true gradient of the loss function while training a VGG-16 network on the CIFAR-10 dataset. We have mentioned this observation in referring to Fig 1 in light of Theorems 3.1 and 3.2 to strengthen our claims and indicate that our theoretical results support empirical observations from deep network training (last line of Section 3.2).
>
> Given these modifications to the manuscript, we believe that we were able to address the concerns raised by the reviewer and were able to improve the overall quality of our work. Subsequently, we hope that the reviewer finds our work to be worthy of acceptance at ICLR. We thank them for their constructive review.

---

> ### Author Response · Authors · 2022-12-11
> **Follow-up on response**
>
> Dear Reviewer Z5BT,
>
> As the discussion period is ending soon, we were wondering if you had any further questions or comments to which we can respond. We would be interested to know if our response addressed your concerns about our manuscript, specifically if our response clarified the questions you had about our work. If so, we would be extremely grateful if you could consider revising your score to reflect the same.
>
> Thank you,
> Authors

---

### Author Response · Authors · 2022-11-18
**Manuscript and Supplementary materials updated**

Dear Reviewers,

We sincerely thank all of you for reading our submission and providing astute and extremely helpful comments to improve the paper. We have made the changes to our manuscript, as outlined in the responses to each your comments. The updated manuscript and supplementary materials (containing the code) are now available for review and comments. Please note that we have also added a Reprodubility Statement at the end of the main text, to indicate that the experimental details are provided in the Appendix and the code is part of the attached supplementary material.

We hope that you find the updated version of our work to be of acceptance quality at ICLR. We are happy to engage in further discussions and provide any clarifications about our work.

Regards,\
Authors

---

### Decision · Program_Chairs · 2023-01-20

**Decision:**

Accept: poster

**Justification For Why Not Higher Score:**

Albeit a good work, it falls short of giving a clear case where gradient estimation involves white noise in implementations for neuromorphic chips, the opening motivation of the paper.

**Justification For Why Not Lower Score:**

The work constitutes a good contribution toward the understanding of gradient estimation.

**Metareview: Summary, Strengths And Weaknesses:**

This paper examines how variance and bias in gradient estimators can affect learning depending on properties such as network size and activity sparsity, the gradient's norm, and the curvature of the loss landscape. The authors show that while variance and bias can impair learning on training data, they may also be beneficial for generalization. They suggest that researchers analyze these properties to determine if their approximation to gradient descent will be effective for learning given their network's task properties.

The reviewers appreciated the theoretical and empirical support provided for the claims. The paper is of high quality, giving a clear value and message, meeting the bar of this venue. Hence, I recommend accepting this paper. I strongly encourage the authors to strengthen the paper's motivation and connect how the gradient estimation is related to implementations with neuromorphic chips. Concrete examples of how true gradients cannot be calculated in those implementations would help readers appreciate the relevance and importance of this work. Moreover, the authors should discuss the relevance of studying the effect of fixed bias and white noise in the context of methods where they occur.




**Note From Pc:**

if the above contains the word "oral" or "spotlight" please see: "oral" presentation means -> notable-top-5% and "spotlight" means -> notable-top-25%. As stated in our emails, we are disassociating presentation type from AC recommendations